# The Mechanism of Manganese Ferrite Nanomaterials Promoting Drought Resistance in Rice

**DOI:** 10.3390/nano13091484

**Published:** 2023-04-26

**Authors:** Le Yue, Budiao Xie, Xuesong Cao, Feiran Chen, Chuanxi Wang, Zhenggao Xiao, Liya Jiao, Zhenyu Wang

**Affiliations:** 1Institute of Environmental Processes and Pollution Control and School of Environment and Civil Engineering, Jiangnan University, Wuxi 214122, China; leyue@jiangnan.edu.cn (L.Y.);; 2Jiangsu Engineering Laboratory for Biomass Energy and Carbon Reduction Technology, Wuxi 214122, China

**Keywords:** nanomaterials, drought resistance, rice, signal transduction, root angle

## Abstract

Strategies to reduce the risk of drought damage are urgently needed as intensified climate change threatens agricultural production. One potential strategy was using nanomaterials (NMs) to enhance plant resistance by regulating various physiological and biochemical processes. In the present study, 10 mg kg^−1^ manganese ferrite (MnFe_2_O_4_) NMs had the optimal enhancement to elevate the levels of biomass, photosynthesis, nutrient elements, and polysaccharide in rice by 10.9–525.0%, respectively, under drought stress. The MnFe_2_O_4_ NMs were internalized by rice plants, which provided the possibility for rice to better cope with drought. Furthermore, as compared with drought control and equivalent ion control, the introduction of MnFe_2_O_4_ NMs into the roots significantly upregulated the drought-sensing gene *CLE25* (29.4%) and the receptor gene *NCED3* (59.9%). This activation stimulated downstream abscisic acid, proline, malondialdehyde, and wax biosynthesis by 23.3%, 38.9%, 7.2%, and 26.2%, respectively. In addition, 10 mg·kg^−1^ MnFe_2_O_4_ NMs significantly upregulated the relative expressions of *OR1*, *AUX2*, *AUX3*, *PIN1a*, and *PIN2*, and increased IAA content significantly, resulting in an enlarged root angle and a deeper and denser root to help the plant withstand drought stresses. The nutritional quality of rice grains was also improved. Our study provides crucial insight for developing nano-enabled strategies to improve crop productivity and resilience to climate change.

## 1. Introduction

The global population is forecast to reach 9.7 billion by 2070 [1]. It is urgent that the challenges of food shortages and excessive population pressure are met. However, climate change is leading us to a hotter and drier world, causing more than 30% of crop losses in the world’s food production process and a USD 30 billion global economic loss over the past decade [2]. Drought, one of the most widespread and damaging environmental stresses, is the major limitation for crop production.

Rice feeds more than 50% of the global population. According to statistics from the Organization for Economic Co-operation and Development (OECD), rice production in 2020 will reach 520.1 billion tons, accounting for 19% of the global production of five major food crops [3]. Rice is more susceptible to drought stress owing to its shallower roots than other crops. Numerous studies indicated that drought could reduce the water potential of rice leaves, accelerate chlorophyll decomposition, decrease photosynthetic rate, and cause a large number of reactive oxygen species (ROS) accumulations, leading to cell membrane damage and ultimately grain yield reduction [4,5]. Meanwhile, a series of resistance and avoidance mechanisms are adopted by plants to resist drought stress [6]. Plants have to adjust accordingly to respond to drought by removing ROS, increasing osmotic pressure, and establishing dehydration tolerance. Rice plants generate more oxidative damage substances due to water shortages and increase the activity of antioxidant enzymes to eliminate this oxidative damage and synthesize more proline to maintain homeostasis in a hyperosmotic state [7]. They also avoid drought-induced stress by closing stomata, curling leaves, increasing root angle, and increasing the activity of aquaporins to reduce water loss or enhance water absorption [8,9].

When plants suffer from drought stress, the root system first senses the lack of water in the external environment through changes in water potential, osmotic pressure, and other physical characteristics [10]. The root-derived CLE25 peptide moves from the roots to the shoots, binding to *NCED* in the stem or leaves to promote stomatal closure by triggering abscisic acid (ABA) accumulation and then enhancing drought resistance [11]. The increase in ABA content in leaves can activate the defense mechanisms of plants under drought stress, such as promoting the biosynthesis of proline and other substances to alleviate the osmotic pressure induced by water scarcity. Meanwhile, deep roots could help plants avoid drought stress by absorbing water from the deep soil layer. Uga et al. (2013) introduced *DRO1*, which controls root angle, into shallow-rooted rice and found that the seed setting rate of the rice under severe drought was increased by up to 30% with the introduction of *DRO1* [12]. As a homolog of *DRO1*, *OR1* is expressed in root columnar cells [13]. Kitomi et al. (2020) found that rice carrying *OR1* would show a deeper root growth phenotype and suffer more significant salt stress in saline-alkali soil, thus reducing rice yield [14]. The AUX and PIN gene families are also closely involved in the indirect regulation of root angle and root hair development by mediating IAA synthesis and transport in root cells to better cope with drought [15].

Recently, nanomaterials (NMs) have been used to gradually release nutrients while minimizing soil pollution. They showed higher efficiency and lower inputs than traditional agrichemicals due to their unique physicochemical properties. The production of nanoscale farming was reported to be 20% higher than that of non-nanoscale farming [16]. The possible roles of NMs in boosting plant defense responses to abiotic stresses have been elucidated. For example, foliar application of 5 mg·L^−1^ carbon dots scavenged ROS in soybean leaves under drought stress, thus improving photosynthesis and carbohydrate allocation [17]. Iron (Fe) NMs enhanced plasma membrane H^+^-ATPase activation and stomatal opening in *Arabidopsis thaliana* under water stress [18]. ZnO NMs in low doses speed up panicle initiation and increase grain yield in wheat grown in arid soil [19]. However, the mechanism underlying the enhancement of rice root drought resistance by NMs remains poorly understood.

The drought could be mitigated by soil amendment or fertilization to improve soil moisture and rice productivity [19]. Fe and manganese (Mn) are essential plant mineral nutrients for stimulating hormones, e.g., IAA, and play key roles in multiple physiological processes. Crop stress management using Fe and Mn has been proven to reduce negative effects and contribute to abiotic stress tolerance through activating plant enzymatic antioxidants to scavenge ROS [20]. In addition, we have previously shown that manganese ferrite (MnFe_2_O_4_) NMs served as electron donors, improving photosynthesis efficiency, activating hormone signaling, upregulating expression of flowering genes, increasing tomato root growth, and ultimately total fruit yield much better than equivalent ionic controls and fertilizer controls [21]. Therefore, we hypothesized that the soil application of MnFe_2_O_4_ NMs could stimulate the drought-sensing ability of rice roots, leading to a cascade of responses that affect photosynthesis and root growth and ultimately enhance the ability of rice plants to withstand drought stress. The objectives of the present study were to investigate: (1) the alleviation of MnFe_2_O_4_ NMs on the growth inhibition caused by drought; (2) the alteration of plant signal transduction under drought responding to MnFe_2_O_4_ NMs; (3) the regulation of root angle by MnFe_2_O_4_ NMs and the related mechanism; and (4) the ultimate impact of MnFe_2_O_4_ NMs on rice yield and quality under drought conditions. This study provides significant insights into the optimization of nano-enabled agriculture for augmenting stress resistance.

## 2. Materials and Methods

### 2.1. MnFe_2_O_4_ NM Synthesis and Characterization

MnFe_2_O_4_ NMs were synthesized as described in previous research [21]. Briefly, 1.984 g of MnCl_2_·4H_2_O and 5.400 g of FeCl_3_·6H_2_O were mixed with 20 mL of ethylene glycol. The mixture was stirred until complete dissolution was achieved, followed by a dropwise addition of 5 mol·L^−1^ NaOH until the pH reached 11. The solution was further stirred for 10 min until a red-brown color was observed. The resulting mixture was then transferred into a 50 mL Teflon-lined reactor for hydrothermal synthesis at 200 °C for 12 h. The synthesized product was subsequently vacuum dried for 6 h to obtain MnFe_2_O_4_ NMs. A transmission electron microscope (TEM, JEM-2100, Nippon Electronics, Toyko, JPN) was used to observe the size and shape of MnFe_2_O_4_ NMs. The zeta potential and hydrodynamic diameter of MnFe_2_O_4_ NMs were analyzed by a Zetasizer (Nano-ZS90, Malvern Instruments, Malvern, UK). The different phases and magnetic behavior of MnFe_2_O_4_ NMs were identified by X-ray diffraction (XRD, D8 Advance, AXS, Berlin, GER) and vibrating sample magnetometer (VSM, PPMS-9, Quantum Design, Columbus, OH, USA), respectively.

### 2.2. Plant Cultivation and NM Exposure

Rice (chuang liang you 669) seeds were purchased from Anhui Lvyi Seed Industry Co., Ltd., Hefei, China. After the seed sterilization, three average-sized seeds were planted in each plastic flowerpot (120 × 178 mm in diameter × height) with 1.5 kg of soil in it. The soil was collected from a farm in MaShan, Wuxi (longitude 120°15′ E and latitude 31°54′ N), with the total nitrogen, the total organic carbon, and the pH of 47.7 g·kg^−1^, 119.7 g·kg^−1^, and 6.8, respectively. All flowerpots were placed in a greenhouse with a light/day photoperiod of 16/8 h, a day/night temperature of 30/25 °C, and a relative humidity of 60 ± 5% at Jiangnan University. MnFe_2_O_4_ NMs were homogenized evenly with paddy soil before rice cultivation. At the beginning of the tillering stage (after growth for 30 days, when the sixth leaf emerged), rice plants were exposed to drought stress (30% moisture in the soil) for 14 days [22]. Five treatment groups were set up in this preliminary experiment, including the non-drought blank control (non-CK), drought control (CK), and rice exposed to both drought and 1, 10, and 50 mg·L^−1^ MnFe_2_O_4_ NMs. Due to the fact that 10 mg·L^−1^ MnFe_2_O_4_ NMs exhibited the best performance to increase the photosynthetic pigment content, photosynthetic products, and plant biomass under drought stress, 10 mg·L^−1^ MnFe_2_O_4_ NMs were selected to explore the subsequent mechanism. An ion control (0.043 mmol·L^−1^ MnSO_4_·H_2_O + FeSO_4_·7H_2_O, which have equivalent Mn and Fe content to that in 10 mg·L^−1^ MnFe_2_O_4_ NMs) was also included. Five biological replicates were set up for each treatment.

### 2.3. Root Morphology, Photosynthesis, Element Content, and Single Particle Concentration

The root morphology (root length, root surface area, root tips, and root volume) was scanned by WinRHIZO Pro 2017 b (Canada). The determination of photosynthetic pigments (chlorophyll a, chlorophyll b, and carotenoids) in rice leaves was based on the research of Rigon et al. (2012) [23]. For the nutrient element determination, a total of 25 mg of rice shoots and roots were digested with a mixture of nitric acid and ultrapure water (*v*/*v* = 1:1) in a microwave-accelerated reaction system at 190 °C and 1400 W (MARS 6, CEM, Matthews, NC, USA). The digested solution was filtered through a 0.22 μm filter membrane before being diluted to 50 mL. The content of elements was then detected by inductively coupled plasma mass spectrometry (ICP-MS, iCAP-TQ, Thermo Fisher, Bremen, Germany).

The determination of single particles of MnFe_2_O_4_ NMs in rice tissues was referred to in previous studies [21]. Briefly, 25 mg of rice shoots and roots were washed five times with DI water and then homogenized. A dose of 50 mg·mL^−1^ pectinase (2 mL) was added to the homogenates prior to shaking for 24 h at 37 °C. After precipitating for 60 min and passing through a 0.45 μm filter membrane, the single particles of MnFe_2_O_4_ NMs in the surface solution were determined by Single Particle-ICP-MS (SP-ICP-MS, Thermo Fisher, Bremen, Germany) analysis.

### 2.4. Determination of IAA and ABA

Forty milligrams of plant tissue were ground into powder in liquid nitrogen, and a mixture of ethyl acetate and BHT-butylhydroxytoluene was added. The suspension was vortexed for 15 min and sonicated in an ice bath for 15 min. After centrifuging at 13,000× *g* for 10 min at 4 °C, the supernatant was gently blow-dried with nitrogen and then redissolved in 200 μL of 70% methanol. After being swirled for 5 min, sonicated at 0 °C for 5 min, and centrifuged at 12,000 rpm at 4 °C, 100 μL of supernatant was obtained to determine the contents of IAA and ABA by HPLC-MS/MS (Thermo Scientific, Germering, Germany).

### 2.5. Quantitative Real-Time PCR (qRT-PCR) Analysis

The expressions of genes (*PIN*, *AUX*, *SWEET*, *SUT*, *DRO*, *OR*, *NCED,* and *CLE*) associated with polar transport of IAA, transport of sugar, root angle, synthesis of ABA, and signal transduction in rice leaves, stems, and roots were measured by qRT-PCR. Sangon Biotech Co., Ltd. (Shanghai, China) designed and synthesized the primers (Appendix A). The total RNA in plant tissues was isolated following an RNA extraction protocol (TaKaRa MiniBEST Plant RNA Extraction Kit) through a T100 Thermal Cycler (Bio-Rad, Hercules, CA, USA). The PCR reaction was conducted in the following order: 10 min at 95 °C, followed by 40 cycles of 95 °C for 15 s, 60 °C for 30 s, and 72 °C for 32 s. The relative expression of each gene was analyzed by the standard calculation 2^−∆∆CT^, and the data was analyzed by Bio-Rad CFX Manager software.

### 2.6. Statistical Analysis

Data for quantification analyses were presented as mean value ± SD. All the measurements were performed at least in triplicate. SPSS25 (IBM SPSS Statistics 25) with a one-way ANOVA with LSD and a *t*-test was used to analyze all data. A *p*-value of less than 0.05 was recognized as a significant difference.

## 3. Results and Discussion

### 3.1. MnFe_2_O_4_ NM Characterization

As shown in Figure 1A, MnFe_2_O_4_ NMs were present in spherical shapes with sizes ranging from 20 to 60 nm. The zeta potential and hydrodynamic diameter of 10 mg·L^−1^ MnFe_2_O_4_ NMs in DI water were −27 mV and 982 nm, respectively (Appendix A). The XRD result (Figure 1B) exhibited that the characteristic peaks of MnFe_2_O_4_ NMs were consistent with the peaks annotated on the standard MnFe_2_O_4_ card (JCPDS card no. 10-0319) [24]. Figure 1C,D showed the Fe and Mn oxidation states and showed that the Fe 2p_3/2_ peak was separated into two peaks: at 710.6 eV (the main peak) and 23.5 eV (the satellite peak). Meanwhile, Fe^3+^ and Fe^2+^ formed Fe 2p_3/2_ peaks at 713.22 eV and 710.6 eV, respectively. Fe 2p_1/2_ also had peaks at 723.49 eV (Fe^2+^) and 726.16 eV (Fe^3+^). Similarly, the Mn 2p_3/2_ peak had two peaks: at 641.17 eV (the main peak) and 645.05 eV (the satellite peak). These satellite structures of Fe and Mn showed that the Fe and Mn ions in MnFe_2_O_4_ NMs were mostly in the 2+ state. The amount of Mn^2+^ (87.33%) was greater than Mn^3+^ (12.67%), and the level of Fe^2+^ (67.42%) was higher than Fe^3+^ (32.58%). Thus, the MnFe_2_O_4_ NMs were not very stable, possibly slowly releasing Mn and Fe ions [21]. The VSM result (Appendix A) showed that MnFe_2_O_4_ NMs has superparamagnetic properties with a saturation magnetization of Ms~60 emu/g.

### 3.2. MnFe_2_O_4_ NMs Alleviated the Adverse Effects of Drought on Rice Growth

The drought significantly reduced the growth of rice shoots and roots; Figure 2A showed significant growth retardation and leaf curling of rice in response to the drought. However, soil application of MnFe_2_O_4_ NMs alleviated this growth inhibition. Although still lower than the non-drought control, the shoot biomass was significantly increased by 77.2% and 70.3% in rice by 1 mg kg^−1^ and 10 mg kg^−1^ MnFe_2_O_4_ NMs over drought control (Figure 2B). As shown in Figure 2C,D, the intercellular CO_2_ concentration and net photosynthetic rate were best elevated by 26.9% and 23.1% upon exposure to 10 mg kg^−1^ MnFe_2_O_4_ NMs, respectively. The enhanced photosynthesis provides more nutrients for plant growth, and the photosynthetic pigments participate in the absorption and transfer of light energy and the primary photochemical reaction in photosynthesis [25]. Accordingly, the content of total photosynthetic pigment and carotenoid in rice was significantly increased by 10.9% and 26.4% after exposure to 10 mg kg^−1^ MnFe_2_O_4_ NMs (Figure 2E,F). Moreover, the nutrient contents were improved by MnFe_2_O_4_ NMs (Appendix A). Particularly, the levels of phosphorus (P), potassium (K), magnesium (Mg), and sulfide (S) in shoots were significantly increased by 14.1%, 26.5%, 46.0%, and 19.8%, while P and S in roots were increased by 27.2% and 49.8%, respectively. The above-mentioned elements are all closely related to photosynthesis and plant development. For instance, P stores light energy in ATP and forms NADPH by participating in photophosphorylation to enhance photosynthesis and promote plant growth; S affects crop quality by accumulating γ-glutamyl dipeptide [26]; Mg is the central element of photosynthesis, involving electron transport, protein synthesis, and nucleotide metabolism; and K plays a crucial role in maintaining adequate stimulation and coordination of transporters in signal regulation pathways [27]. As a result, more polysaccharide was accumulated in rice by MnFe_2_O_4_ NMs, and 10 mg kg^−1^ showed the most promotion effect by 329.1% and 525.0%, respectively, in rice shoot and root as compared with the drought control (Figure 2G,H), which would help provide more energy and contribute to better plant growth under drought stresses. Therefore, 10 mg·kg^−1^ MnFe_2_O_4_ NMs was the optimal concentration to alleviate rice growth inhibition under drought stress and was selected to further explore its mechanisms of enhanced resistance to drought.

### 3.3. Accumulation of MnFe_2_O_4_ NMs in Rice Plants under Drought Stress

Figure 3 showed that the contents of Fe and Mn in rice plants after application of 10 mg·kg^−1^ MnFe_2_O_4_ NMs were significantly increased. The contents of Fe and Mn in roots were elevated by 47.7% and 166.3% as compared to the drought control and by 13.2% and 30.9% as compared to the ion control (Figure 3A,B). The contents of Fe and Mn were elevated by 25.3% and 108.9% in shoots over the drought control, respectively. The results of SP-ICP-MS showed that the particle numbers of MnFe_2_O_4_ NMs in rice root and leaf were significantly increased by 47.0% and 41.24%, respectively (Figure 3C). In addition, the magnetic properties of the shoots and roots after exposure to MnFe_2_O_4_ NMs were obviously stronger than those of the drought control and the ion control (Figure 3D,E). These results indicated that MnFe_2_O_4_ NMs might enter rice plants. Once taken up by plants, NMs could be transported and accumulated in different parts, interacting with various cellular components and further impacting plant physiology and metabolism [16]. Fe, a vital element for photosynthesis, facilitates electron transfer processes in Photosystem II, and a series of enzymes involved in photosynthesis rely on Mg as a cofactor [28], which surely strengthens photosynthesis (Figure 2). Additionally, NMs were proven to be more efficient than their equivalent ion control systems because of their unique properties [29]. The internalized NMs might explain the obvious greater enhancement of rice growth after NM treatment than the equivalent ion treatment (Appendix A). Therefore, MnFe_2_O_4_ NMs could enter rice plants, providing the possibility for rice to better cope with drought [29].

### 3.4. MnFe_2_O_4_ NMs Enhanced Signal Transduction in Rice Roots

After soil application of MnFe_2_O_4_ NMs, the expression of the *CLE25* gene was significantly upregulated by 29.4% as compared to the drought control (Figure 4A). As an upstream sensing gene family for drought, CLE activates the resistance of whole plants to drought by synthesizing the small peptide cle25 and transferring it to the shoot [30], where it is accepted by the receptor [11]. The response gene *NCED3* in the shoot with MnFe_2_O_4_ NMs was significantly upregulated by 59.9% compared with the control (Figure 4B), indicating an enhancement of the signal intensity of rice in response to MnFe_2_O_4_ NMs under drought stress. *NCED3* encodes an enzyme that catalyzes the rate-limiting step in ABA synthesis in plants. The ABA content in rice shoots after application of MnFe_2_O_4_ NMs under drought stress was significantly increased by 23.3% compared with the drought control (Figure 4C). ABA has been proven to activate a series of responses; the increase in ABA content could further activate the responses of the shoot to drought stress, such as stomatal closure, cuticle thickening, and proline synthesis, to reduce water loss or attenuate drought-induced oxidative damage [31]. At the same time, the wax synthesized in the leaves was increased by 24.4% compared with the drought control (Figure 4D), suggesting that the cuticle was thickened and that transpiration and water loss could thus be weakened [32]. Furthermore, 10 mg kg^−1^ MnFe_2_O_4_ NMs significantly increased the proline content of rice leaves (Figure 4E) and decreased malondialdehyde (MDA) content (Figure 4F). Compared with the drought control, the level of proline in MnFe_2_O_4_ NMs was significantly increased by 11.7%. As a protective substance under osmotic stress, proline maintains the homeostasis of plants under stress [7]. Compared with the non-drought control, MDA content in rice leaves decreased by 23.3% with MnFe_2_O_4_ NMs, while the ion control group had no significant effect. MDA, which is a secondary end product of lipid peroxidation, can be used as a biomarker for drought resistance [33].

### 3.5. MnFe_2_O_4_ NMs Changed Root Development under Drought Stress

As the photosynthesis of rice plants was significantly enhanced by MnFe_2_O_4_ NMs over drought control (Figure 3), more photosynthetic products would be transported from shoots to roots. *SWEET11* and *SWEET14*, responsible for sugar transport in the SWEET gene family, were significantly upregulated by 86.4% and 67.1% with 10 mg·kg^−1^ MnFe_2_O_4_ NMs, respectively, compared with the drought control (Appendix A). The transport of polysaccharides from shoots to roots provides energy for root growth [34,35]. In addition, the cellulose content in rice roots treated with MnFe_2_O_4_ NMs decreased by 8.1% compared with the control (Appendix A). Kim et al. (2014) found that Fe-based NMs attacked cellulose and pectin between root cells through releasing electrons, which thins out the cell wall, weakens the mechanical resistance of root growth, and promotes root cell elongation [14]. The absorption and release of electrons may be caused by the change of Fe and Mn valence in MnFe_2_O_4_ NMs (Figure 1), which bombards the cellulose between root cells to thin the cell wall and promote root cell elongation. Thus, the promoted transport of photosynthates from shoot to root and the reduced thickness of the cell wall by MnFe_2_O_4_ NMs could contribute to root elongation. Correspondingly, 10 mg·kg^−1^ MnFe_2_O_4_ NMs induced more and larger root tips in rice under drought stress. Compared with the drought control, MnFe_2_O_4_ NMs increased root tip number, surface area, volume, and root length by 75.9%, 35.2%, 29.4%, and 43.5%, respectively (Appendix A). The denser and deeper roots help obtain water from deep soil to cope with water scarcity [36]. Interestingly, larger root angles were induced. The average root angle of rice under drought stress was 64.14° after 10 mg kg^−1^ MnFe_2_O_4_ NM exposure, which was 10.8% higher than that of the drought control. However no significant difference in root angle was found between the ion group and the drought control (Figure 5A,B).

*DRO1* and *OR1* are concentrated in the apical meristem and around the static center of the root cap, regulating root angle. *DRO1* regulates the elongation of different lateral root cells and thus the root angle by mediating differences in the polar transport of IAA by roots under gravity induction [12]. However, no significant changes were observed in *DRO1* expression in roots (Appendix A). While *OR1*, a homologue of *DRO1,* controls root angle, it was significantly upregulated by 14.8% and 17.1% by 10 mg kg^−1^ MnFe_2_O_4_ NMs as compared to the drought control and the iron control, respectively (Figure 5C). This explains the enlarged rice root angles exposed to NMs to avoid drought stress. Moreover, the polar transport of auxin indirectly regulates root angle [12]. Among the AUX family of inflow genes controlling IAA polar transport, *AUX2* and *AUX3* expressions were upregulated by 18.1% and 38.2%, respectively, as compared with the drought control (Figure 5D), and *PIN1a* and *PIN2*, in the PIN family of efflux genes controlling polar transport of IAA, were upregulated by 31.5% and 18.0%, respectively (Figure 5E). As well as the level of IAA, which increased by 74.5% (Figure 5F). Root growth is closely related to IAA. It has been demonstrated that the expressions of AUX and PIN in Arabidopsis roots were upregulated under low phosphorus conditions to promote the downward polar transport of IAA, which resulted in more root hairs absorbing phosphorus from the soil [37]. Above all, under drought conditions, the soil application of MnFe_2_O_4_ NMs upregulated the gene *OR1* and promoted IAA synthesis and transport to increase the root angle and produce more root hairs, which could help rice plants absorb more water in response to drought [14,38,39].

### 3.6. MnFe_2_O_4_ NMs Enhanced the Grain Yield and Quality of Rice under Drought Stress

After the whole growth period, the rice yield decreased significantly under drought stress, as shown in Figure 6A. The grain filling rate of the drought control was reduced by half compared to the non-drought control. However, 10 mg kg^−1^ MnFe_2_O_4_ NMs significantly mitigated this inhibition, increasing the grain filling rate by 61.1% and 46.7% compared with the drought control and equivalent ion control, respectively (Figure 6B). The thousand-grain weight, grain number, and panicle length were also significantly increased by 22.5%, 19.6%, and 41.3% by NMs over the drought control (Figure 6C–E). The main protein components in rice grains include albumin, globulin, gluten, and prolamin. The drought resulted in a significant decrease in albumin and gluten contents in rice grains. Nevertheless, there was no significant difference in globulin, albumin, and prolamin contents among CK, NMs, and ion treatments after application of 10 mg kg^−1^ MnFe_2_O_4_ NMs (Appendix A). Only the content of gluten in rice grains was significantly increased by 15.7% upon NM exposure (Figure 6F). Gluten has a higher nutritional value but also has fewer negative effects on rice taste. It exists in the endosperm of rice grains in the form of a protein body and contains more essential amino acids, such as arginine, lysine, and glycine, as compared with the other three proteins. Further analysis of nutrient elements in rice grains indicated that the drought stress significantly reduced the levels of Ca, P, Mg, and Zn elements by 10.6–57.9%. MnFe_2_O_4_ NMs significantly increased the contents of Ca, P, Fe, Mn, and K elements by 135.1%, 21.2%, 43.6%, 31.1%, and 19.3% over the drought control, and no significant difference was observed between the drought control, NMs, and the ion control in S, Mg, and Zn contents. Therefore, MnFe_2_O_4_ NMs drove the absorption of nutrient elements by rice plants, leading to changes in the content of the corresponding elements in the grain.

## 4. Conclusions

In the present study, the potential of MnFe_2_O_4_ NMs to induce drought resistance in rice plants was assessed. The results showed that the soil application of 1, 10, and 50 mg kg^−1^ MnFe_2_O_4_ NMs alleviated the growth inhibition caused by drought. The optimal enhancement was observed at 10 mg kg^−1^ MnFe_2_O_4_ NMs, which elevated the levels of biomass, photosynthesis, nutrient elements, and accumulated polysaccharide by 10.9–525.0%, respectively, in rice shoot and root compared with the drought control. SP-ICP-MS and VSM analysis confirmed that MnFe_2_O_4_ NMs were internalized in rice plants, suggesting the possibility that rice plants could cope better with drought. Furthermore, as compared with the drought control and the ion control, the introduction of MnFe_2_O_4_ NMs into the root significantly upregulated the drought-sensing gene *CLE25* by 29.4% in rice roots, which promoted the synthesis of more drought-response signal cle25 peptides and transported them to the shoots, bound by the receptor gene *NCED3* (upregulation by 59.9%). This further activated downstream ABA biosynthesis, leading to a significant increase in ABA content of 23.3%. Correspondingly, the content of proline, MDA, and wax in rice leaves increased by 38.9%, 7.2%, and 26.2%, respectively. MnFe_2_O_4_ NMs at 10 mg·kg^−1^ also significantly increased the root angle of rice, resulting in deeper and denser roots. This was due to the upregulation of the root angle-regulated gene *OR1* by 14.8%, as well as the expression of *AUX2*, *AUX3*, *PIN1a*, and *PIN2*, which promote IAA transport from the root tip to the meristem and increase root development. Moreover, MnFe_2_O_4_ NMs enhanced the biosynthesis of IAA in roots by 23.3% and 14.2% than drought control and ion control, respectively. The gene regulation and IAA content induced by MnFe_2_O_4_ NMs would help rice be better resistant to drought stress. In addition, 10 mg·kg^−1^ MnFe_2_O_4_ NMs increased grain filling rate, thousand grain weight, and panicle length by 61.1%, 22.5%, and 41.3%, respectively. The nutritional quality of rice grains was improved to a certain extent. Overall, the soil application of MnFe_2_O_4_ NMs can not only alleviate the inhibition caused by drought on rice growth but also promote rice yield and improve the quality and flavor of rice grains under drought stress. This study provides significant insight for developing strategies to improve crop productivity and resilience to climate change.

## Figures and Tables

**Figure 1 nanomaterials-13-01484-f001:**
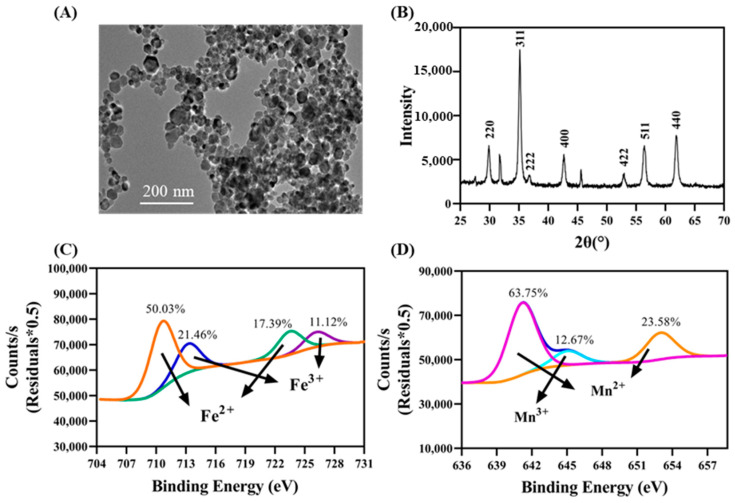
MnFe_2_O_4_ NM characterization. (**A**) TEM image; (**B**) XRD analysis; and (**C**,**D**) XPS results.

**Figure 2 nanomaterials-13-01484-f002:**
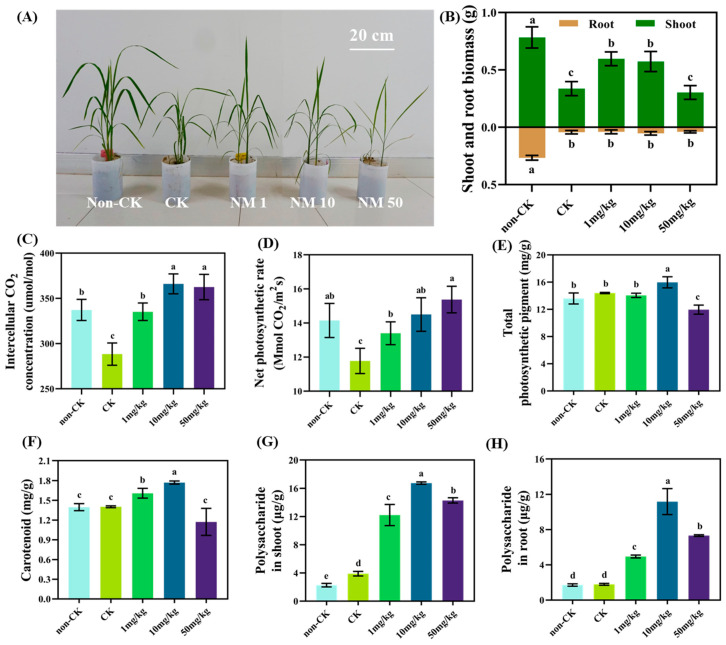
The growth of rice plants under drought conditions in response to different concentrations of MnFe_2_O_4_ NMs. (**A**) Growth phenotype; (**B**) biomass; (**C**) intercellular CO_2_ concentration; (**D**) net photosynthetic rate; (**E**) total photosynthetic pigment in leaves; (**F**) carotenoid content in leaves; (**G**) shoot; and (**H**) root polysaccharide content. The values are given as mean value ± standard deviation (n = 5). Different letters mean significant differences (*p* < 0.05).

**Figure 3 nanomaterials-13-01484-f003:**
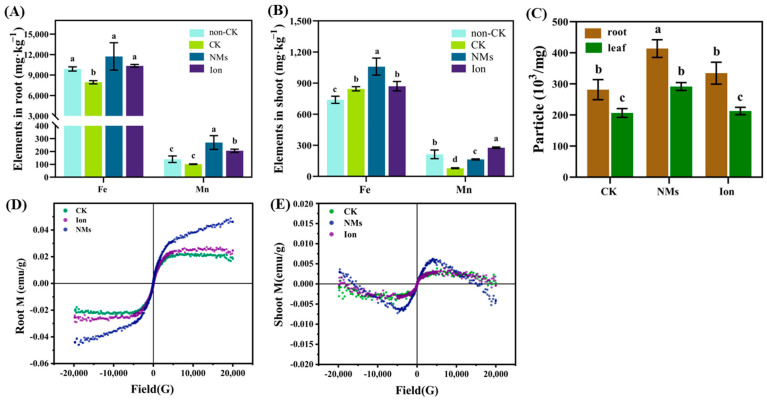
Element content in rice roots (**A**) and shoots (**B**), particle concentration (**C**), and VSM curves of roots (**D**) and shoots (**E**) upon MnFe_2_O_4_ NM exposure as compared to the drought control and the ion control. The values are given as mean value ± standard deviation (n = 5). Different letters mean significant differences (*p* < 0.05).

**Figure 4 nanomaterials-13-01484-f004:**
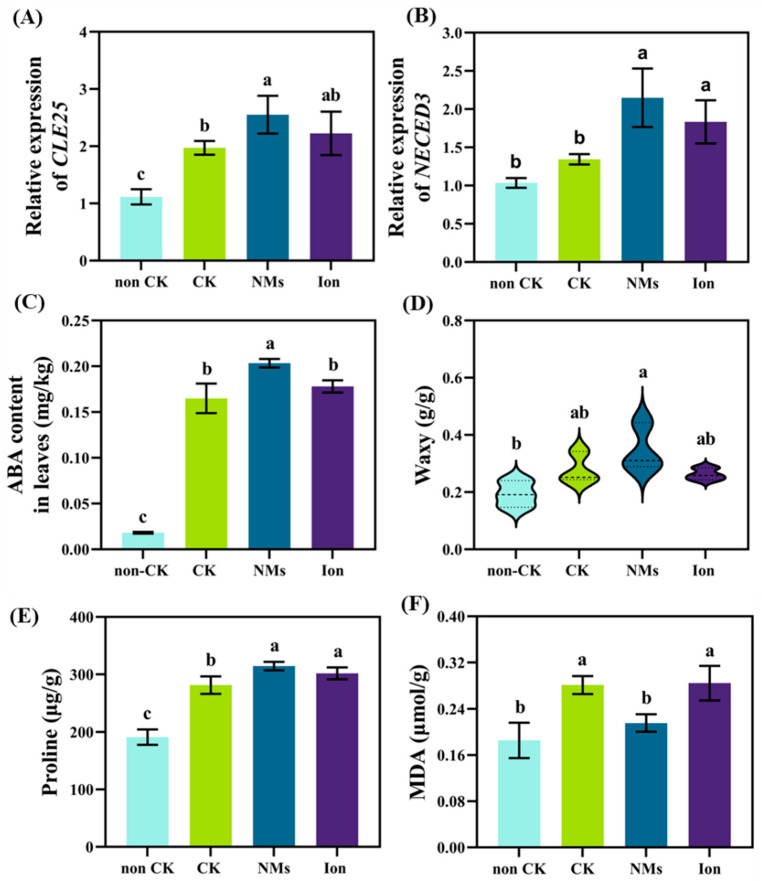
The signal molecule of rice plants responds to 10 mg·kg^−1^ MnFe_2_O_4_ NMs and equivalent ion control under drought. The relative expressions of *CLE25* (**A**) and *NCED3* (**B**) and the content of ABA (**C**), wax content (**D**), proline (**E**), and MDA (**F**). The values are given as mean value ± standard deviation (n = 5). Different letters mean significant differences (*p* < 0.05).

**Figure 5 nanomaterials-13-01484-f005:**
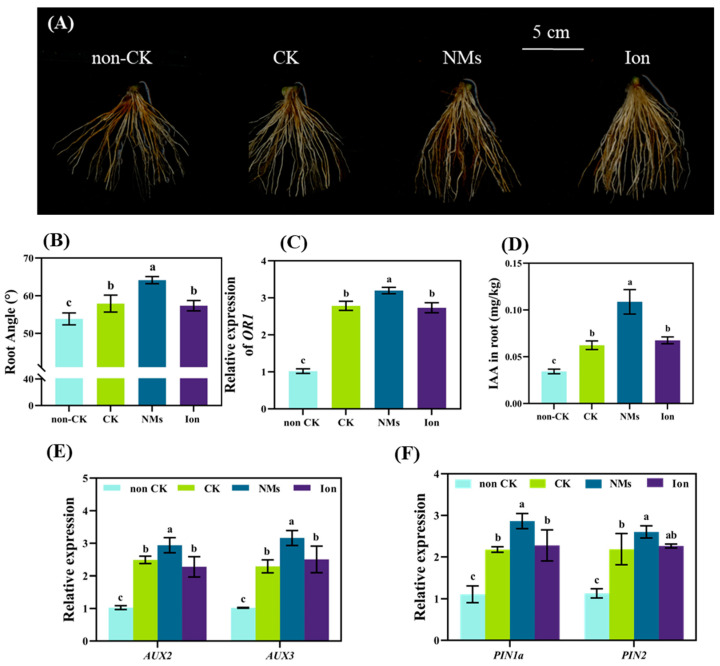
Enhanced rice root angle regulated by MnFe_2_O_4_ NM exposure and equivalent ion control under drought: (**A**) root photos; (**B**) root angle; (**C**) the relative expression of *OR1*; (**D**) IAA content; and the relative expressions of AUX (**E**); and PIN (**F**). The values are given as mean value ± standard deviation (n = 5). Different letters mean significant difference (*p* < 0.05).

**Figure 6 nanomaterials-13-01484-f006:**
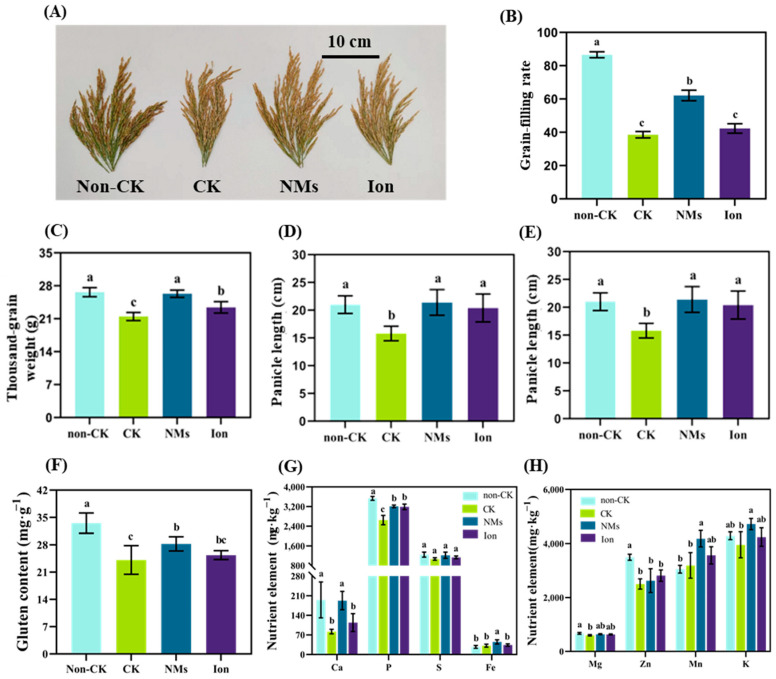
Rice yield and quality after exposure to 10 mg kg^−1^ MnFe_2_O_4_ NMs and equivalent ion control under drought: (**A**) rice spike; (**B**) grain filling rate; (**C**) thousand grain weight; (**D**) grain numbers; (**E**) panicle length; (**F**) gluten content; (**G**) content of Ca, P, S, and Fe; and (**H**) content of Mg, Zn, Mn, and K. Values represent the mean ± standard error (n = 5). Differences between the control and treatment (*p* < 0.05).

## Data Availability

The data presented in this study are available on request from the corresponding author.

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
