# Peer review of "The Mechanism of Manganese Ferrite Nanomaterials Promoting Drought Resistance in Rice"

_nanomaterials, 2023, doi:10.3390/nano13091484_

Round 1
Reviewer 1 Report
Nanomaterials] Manuscript ID: nanomaterials-2364800
Type of manuscript: Article
Manuscript Title: The Mechanism of Manganese Ferrite Nanomaterials Promoting Drought Resistance in Rice
The manuscript reports an interesting experimental study on drought resistance in rice plants and roots (under drought stress) relived by manganese ferrite (MnFe2O4) nanomaterials.
The manuscript is well written and organised, tas well as he citations are complete.
I only note two points below and I’ am in favour of acceptance of the manuscript as it is in Nanomaterials, after, just, including the suggestions below mentioned.
the authors briefly report the synthesis of nanoparticles, as described in Ref. 21, instead of citing only Ref. 21.
The VSM curves of roof of Figure 3(D) and 3(E)are, in my opinion, are quite impressive.
The authors report on the potential toxicity of rice subjected to Manganese ferrite nanoparticles.
English is fine
Reviewer 2 Report
The paper reports on the positive effect of manganese ferrite nanoparticles on the drought resistnce in rice and on the molecular mechanisms underlying this effect. The subject of the paper fits well the scope of Nanomaterials journal. The results presented are new and worth of being published.
I have the following comments:
1. My major concern is that the action mechanism of nanoparticles was not disclosed. Are they dissolved partly or completely in the plant sap? Or do they remain unchanged? In other words, do MnFe2O4 act as a nanozyme or are they dissolved to produce free iron and manganese ions? Please explain.
2. Some peaks in the XRD pattern shown in Fig. 1 remain unattributed. Are they associated with any admixtures?
3. No particle size distribution is provided in Fig. 1 (see figure caption).
4. Please check the formatting and font size throughout the manuscript.
The quality of English is good enough.
